# A potential photo-protective, antioxidant function for DMSO in marine phytoplankton

**Brandon J. McNabb** [ID] [1]*, **Philippe D. Tortell** [1,2]

**1** Department of Earth, Ocean and Atmospheric Sciences, University of British Columbia, Vancouver, BC, Canada, **2** Department of Botany, University of British Columbia, Vancouver, BC, Canada

* bmcnabb@eoas.ubc.ca

**Data Availability Statement:** Experimental data supporting the findings of this study are available at the SEANOE repository "DMS/O/P cycling in coastal NE subarctic Pacific phytoplankton assemblages exposed to elevated irradiance and

## Abstract

The marine compound dimethyl sulfoxide (DMSO) is ubiquitous in the world's surface ocean, constituting one of the largest sources of reduced organic sulfur in seawater. DMSO cycling has been linked to the formation of the climate-active gas dimethyl sulfide (DMS) through a reductive pathway, but the underlying physiological role of DMSO reduction, and the environmental controls on this pathway, remain unresolved. Here we present evidence that DMSO reduction to DMS serves an antioxidant role in phytoplankton through a secondary electron-scavenging pathway that can dissipate excess light-harvested energy, and potentially mitigate the formation of reactive oxygen species (ROS). Results from isotopic tracer experiments demonstrate significant increases in DMSO reduction rates in low-light acclimated natural phytoplankton assemblages exposed to high irradiance. Increased DMSO reduction rates were negatively correlated with non-photochemical quenching, while treatment with the photosynthetic electron transport inhibitor DCMU significantly decreased DMSO reduction, indicating a link to photosynthetically-derived electrons. Our results show that photic stress drives enhanced DMSO reduction in marine phytoplankton, linking DMS production to irradiance and vertical mixing through an electron scavenging mechanism that could serve an antioxidant role.

## Introduction

The biogenic compound dimethyl sulfide (DMS) is crucial to the global marine sulfur cycle and microbial ecology. DMS influences marine food webs as a chemical foraging cue for predators [1,2], and can satisfy metabolic carbon [3] and sulfur [4] demands in certain microbial taxa. DMS is rapidly ventilated from the surface ocean, contributing ~27 Tg S yr$^{-1}$ to the atmosphere globally [5]. The oxidation products of DMS yield cloud-nucleating aerosols that can increase regional albedo and affect the atmospheric radiative budget [6–8].

The oceanic distribution of DMS is controlled by complex abiotic and biological cycling, involving several production and consumption pathways. The enzymatic cleavage of the algal metabolite dimethyl sulfoniopropionate (DMSP) to DMS has traditionally been considered as the primary production pathway for DMS. This cleavage pathway accounts for a significant fraction of bacterial and algal DMSP turnover [9,10], although many phytoplankton species

photosynthetic inhibition (2022-2023)" (DOI: 10.17882/101468). Ancillary CTD data used to derive physical water properties (i.e. MLD, Zeu) from cruises PAC 2022-022 and PAC 2023-026 are publicly available at https://www.waterproperties.ca.

**Funding:** "This work was supported by grants awarded to P.D.T. and B.J.M through the National Science and Engineering Research Council (NSERC). The funders had no role in study design, data collection and analysis, decision to publish, or preparation of the manuscript."

**Competing interests:** The authors declare no competing interests.

lack the lyase enzyme needed for DMSP cleavage [11]. Emerging evidence suggests that DMSO reduction is also a significant source of DMS, with yields from this pathway occasionally equal to, or even exceeding, the yield from DMSP cleavage [12–14]. Although concentrations of DMSO can be very high in seawater [15], the distribution and cycling of this compound have received far less attention than DMS and DMSP. Recent work has shown that DMSO can be produced by both marine bacteria and phytoplankton through the coupled oxidation-cleavage of DMSP involving the intermediate dimethylsulfoxonium propionate (DMSOP) [16]. DMSO formation also contributes significantly to DMS loss in the surface ocean through both microbial oxidation [3] and abiotic photooxidation [17] of DMS.

Increasing evidence suggests that the biological pathways of DMS/P/O cycling are at least partially stress-driven, with elevated concentrations and turnover of these compounds observed under conditions that induce physiological stress [10,12,18–20]. Although the exact mechanism driving this cycling remains unclear, one prominent hypothesis is that DMSP, DMS and DMSO can act as antioxidants by scavenging reactive oxygen species (ROS) [18]. More recent work suggests that DMS is less likely to function as an intracellular antioxidant due to its high membrane permeability, although it may still function in scavenging ROS within the cellular membranes of species with high DMSP synthesis and cleavage activity [21]. High irradiance is a potential oxidative stressor, generating ROS through oversaturation of photosynthetic redox carriers and potential photosystem II (PSII) damage [22,23]. In the oceans, the irradiance received by phytoplankton is modulated by vertical mixing [24], and both models and observations have shown that irradiance and mixing regimes can significantly influence global DMS distributions [25–27].

While DMSO turnover has been assumed to result primarily from oxidative scavenging of ROS [18], less is known about the DMS-forming DMSO reduction pathway and its potential to alleviate photo-oxidative stress. DMSO reduction is nearly ubiquitous in marine bacteria and within several classes of phytoplankton [14]. This pathway is upregulated in response to nitrogen and vitamin $B_{12}$ limitation in phytoplankton, providing indirect evidence that it serves an antioxidant role [28]. Recent observations have also demonstrated decreased DMSO concentrations during periods of mid-day high irradiance when non-photochemical quenching mechanisms were induced. This result further suggests an enhancement of DMSO reduction under conditions of potential oxidative stress [12]. Current evidence linking DMSO reduction to photo-oxidative stress remains correlational, and no experimental studies have directly explored this potential relationship within natural plankton assemblages. To address this knowledge gap, we conducted field experiments to investigate the dynamics of DMSO reduction and other DMS cycling pathways in response to changes in potential photo-oxidative stress and photosynthetic electron transport.

## Materials and methods

### Study area and oceanographic conditions

Field experiments were conducted on board the *CCGS John P Tully* between August 25[th], and September 4[th], 2022 (cruise ID: PAC 2022–022), and between August 24 and September 5[th], 2023 (cruise ID: PAC 2023–026). Sampling stations were chosen along the west and north coast of Vancouver Island, British Columbia, targeting phytoplankton assemblages across the continental shelf break with varying depths of the subsurface chlorophyll-a (chl-a) maximum (Figs 1D and S1). Mixed layer depth (MLD) was calculated as the depth at which seawater density increased by 0.125 kg m$^{-3}$ relative to the surface [27]. Seawater density was calculated from conservative temperature and absolute salinity derived from the Python implementation of the TEOS-10 toolbox (python-gsw package, v.3.6.17). The euphotic zone depth ($Z_{eu}$) was

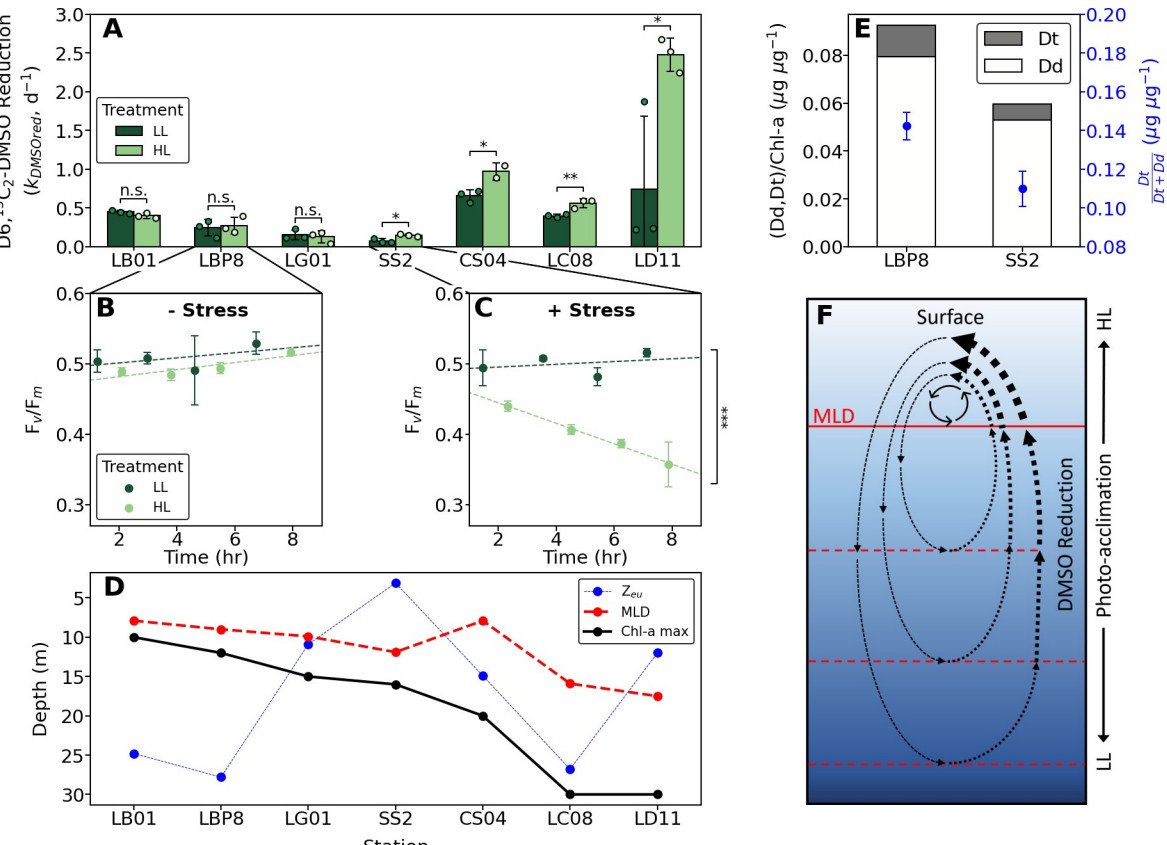

**Fig 1. DMSO reduction in response to high irradiance exposure from simulated vertical mixing. (A)** D6,$^{13}$C$_2$-DMSO reduction rate constants ($k_{DMSOred}$, in d$^{-1}$) in the low light (LL) and high light (HL) treatments across all stations. Data points represent individual replicate measurements, while mean rates (denoted by bars) are derived from n = 3 replicates, except for LBP8—LL and CS04—HL (n = 2). Significance is determined by two-tailed Student's t-tests (*: p < 0.05, **: p < 0.01, n.s.: non-significant). **(B, C)** Changes in maximum photosynthetic efficiency ($F_v/F_m$, n = 3) under LL and HL exposure for stations LBP8 and SS2, respectively. Significant simple main effects between treatments were found for SS2 (two-way mixed effects ANOVA model; ***: p < 0.001). Dashed lines represent linear regression fits over time. **(D)** Sampling depth at the chlorophyll-a maximum (Chl-a max), euphotic zone depth ($Z_{eu}$), and mixed layer depth (MLD) for each station. **(E)** Chl-a normalized concentrations of xanthophylls diadinoxanthin (Dd) and diatoxanthin (Dt) (µg µg$^{-1}$; bars, n = 2) and their de-epoxidation ratios (blue markers, n = 2) for stations LBP8 and SS2. Higher concentrations and de-epoxidation ratios of Dd and Dt are indicative of increased photoprotection through thermal dissipation [29]. All error bars denote range (n = 2) or ± 1 s.d. (n = 3). **(F)** Conceptual schematic illustrating the hypothesized link between vertical mixing (dashed arrows), the light exposure history of the natural assemblages, and DMSO reduction rates (symbolized by the dashed arrow thickness). Processes shoaling the MLD (denoted by red dashed lines) will entrain sub-surface, LL-acclimated assemblages to higher irradiance at the surface, promoting increased DMSO reduction rates (thicker dashed arrows).

calculated as the depth of 1% surface photosynthetically active radiation (PAR) from downcast PAR profiles. Ancillary data available from these experiments are summarized in S2 Table.

## Incubation experiments

Incubation experiments were conducted with isotopically-labelled DMS, DMSP and DMSO to quantify the simultaneous turnover of these molecules by plankton assemblages [13,30]. Daily working stocks of these tracers were prepared immediately prior to the start of each incubation, and triplicate samples were spiked with tracer concentrations targeting ~10% of ambient concentrations, corresponding to 0.5 nM deuterated D3-DMS (CDN Isotopes, 99% purity), 2 nM deuterated D6-DMSP (prepared with the method of Challenger and Simpson [31]) and 2 nM deuterated, $^{13}$C-labelled D6,$^{13}$C$_2$-DMSO (Sigma Aldrich, 99% purity). Actual

concentrations of D6,$^{13}C_2$-DMSO reached up to 5.6 nM in some experiments. The D6-DMSP stock was pre-sparged for $\geq 1$ hour to remove endogenous D6-DMS prior to incubations. Some experimental replicates were treated with the inhibitor (3-(3,4-dichlorophenyl)-1,1-dimethylurea; DCMU) to block photosynthetic electron transport downstream of the secondary quinone acceptor ($Q_B$). Working stock solutions of DCMU (2 μM) were prepared prior to each voyage by dissolving the inhibitor (commercially labelled as Diuron; Sigma Aldrich, $\geq 98\%$ purity) first in acetone, then diluting $\geq 1000$ times into deionized water, with a final two-hour heating step at 60°C to remove the remaining acetone before sealing the solutions in polycarbonate bottles [32,33].

To initiate tracer experiments at each station, a 10 L carboy was filled from a Niskin bottle sampled at the chl-a maximum. The carboys were kept shielded during filling to minimize ambient light exposure and held for up to 6 hours in the dark at 4°C prior to the start of experiments. The carboys were gently homogenized, and 1 L aliquots were transferred into UV transparent FEP bags (Welch Fluorocarbon) equipped with Luer Lock sampling ports. Each bag was then spiked with the sulfur tracers, and for the DCMU experiments, each treatment replicate was additionally spiked with 10 nM DCMU to induce partial inhibition of PSII [34]. The bags were sealed to remove the headspace, gently homogenized, and then placed inside a deck-board incubator kept near sea surface temperature. Incubation experiments were run over ~8–10 hours during daylight hours (between approximately 8 am and 7 pm). For light manipulation experiments, the incubation bags were held inside acrylic tubes wrapped with neutral density screening to restrict irradiance to either ~50% or ~1% of surface levels for the HL and LL treatments, respectively. For DCMU experiments, both control and DCMU-spiked replicates were placed directly inside the incubator, which was wrapped with neutral density screening corresponding to HL treatment. We note that irradiance in our experiments consisted of primarily PAR wavelengths (400–700 nm) due to attenuation of ultraviolet wavelengths by both the incubator lid and acrylic tubes. The lack of ultraviolet (UV) wavelengths may have reduced both abiotic photo-oxidation and biological oxidation of the tracers [23,35,36]. The tracers were allowed to equilibrate inside the bags for one hour prior to the start of sampling, after which 5 mL samples were taken hourly with gas-tight glass syringes with Luer Lock fittings (Fortuna). Treatments were sampled on alternating hours to reduce batch sizes and minimize benchtop exposure of samples. Ancillary data collected for the light manipulation and DCMU incubation experiments are summarized in S2 Table.

## Quantification of DMS and DMSO

DMS and DMSO samples were quantified on an atmospheric pressure chemical ionization mass spectrometer (SCIEX API 3200-series) coupled to a custom-built purge-and-trap gas extraction system [37]. The instrument was configured in multiple reaction monitoring (MRM) mode to ensure high selectivity for native and isotopically-labelled DMS. Chemical ionization is achieved through proton transfer ($[M + H^+]$) in this instrument, yielding mass-to-charge ratios (m/z) offset by +1 unit from their native m/z of each DMS tracer. These m/z were 63 m/z (unlabelled DMS), 66 m/z (D3-DMS), 69 m/z (D6-DMS, derived from D6-DMSP cleavage), and 71 m/z (D6,$^{13}C_2$-DMS, derived from D6,$^{13}C_2$-DMSO reduction). Daily triplicate standards were run to ensure satisfactory performance. Detection limits (calculated as $\bar{x}_{blanks} + 3\sigma_{blanks}$, n = 4) ranged from 0.002 to 0.04 nM for the isotopically labelled tracers. To quantify rates of DMS and DMSP oxidation to DMSO, DMSO samples from station SS5 were flash frozen and preserved at -80°C until subsequent analysis in the laboratory using electrochemical reduction to DMS as described in McCulloch & Tortell [38].

## Sulfur turnover rate constants

All rate constants ($k$, d$^{-1}$) were calculated from the slope of linear regression of the DMS or DMSO concentrations over 3 to 5 time points [13]. These $k$ values (S1 Table) represent gross D3-DMS consumption (D3-DMS loss, 66 m/z), net D6-DMSP cleavage (D6-DMS yields, 69 m/z), net D6,$^{13}$C$_2$-DMSO reduction (D6,$^{13}$C$_2$-DMS yields, 71 m/z), net turnover of unlabelled DMS (63 m/z), and net oxidation of D3-DMS or D6-DMSP (D3-DMSO and D6-DMSO yields; 66 m/z and 69 m/z, respectively). We set a detection limit for these $k$ values, based on the goodness of fit of linear regressions through the time-course data (R$^2$ ≥ 0.5) [12,39].

## Diagnostic pigments

Concentrations of taxonomic marker pigments, xanthophylls, and other photoprotective pigments were measured by high performance liquid chromatography (HPLC) ([40], and references within). At each station, duplicate 1 L aliquots were subsampled from the carboy prior to incubation and filtered through a glass fiber filter (GFF, nominal pore size 0.7 μm), with samples shielded to minimize ambient light exposure during filtration. The filters were flash frozen and stored at -80˚C until subsequent HPLC analysis in the laboratory. The pigments diadinoxanthin (Dd) and diatoxanthin (Dt) are involved in primary photoprotection through non-photochemical quenching (NPQ) pathways, and their intracellular concentrations relative to chlorophyll-a are regulated in response to the photo-acclimative state of phytoplankton [29,41]. Thus, chlorophyll-a normalized concentrations of these pigments (μg L$^{-1}$) and their de-epoxidation ratios ([Dt] / [Dd + Dt]; μg μg$^{-1}$) have been used as markers of photo-acclimation and active xanthophyll cycling associated with NPQ, respectively [42,43]. Phytoplankton taxonomic composition was derived using CHEMTAX (v.1.95) [44] analysis of pigment concentrations, with an initial pigment ratio matrix tuned for NESAP phytoplankton assemblages [45,46]. CHEMTAX results were obtained from the mean of nine runs, each with 20 sets of randomized pigment ratios input into the software.

## Fluorescence-based photo-physiology

Measurements of the maximum quantum yield of photochemistry for PSII ($F_v/F_m$) and non-photochemical quenching (NPQ$_{sv}$) were quantified by Fast Repetition Rate fluorometry (FRRf) using a Light Induced Fluorescence Transient (LIFT) fast repetition rate fluorometer (LIFT-FRRf; Soliense Inc.). For this analysis, samples were dark acclimated for 20 mins, followed by a 1-min exposure to 5 μmol quanta m$^{-2}$ s$^{-1}$ of LED light at 750 nm light to promote NPQ relaxation and alleviate fluorescence quenching associated with electron backflow to PSII [47,48]. Immediately following dark acclimated measurements, the samples were exposed to saturating light by a 1-min exposure to 150 μmol quanta m$^{-2}$ s$^{-1}$ (consisting of 30 μmol quanta m$^{-2}$ s$^{-1}$ from each of five LEDs with peak wavelength excitation corresponding to 445, 470, 505, 535, and 590 nm). This light level corresponds to the upper range of saturating irradiance reported previously for PSII photochemistry within NESAP natural assemblages (*i.e.* ~100–150 μmol quanta m$^{-2}$ s$^{-1}$) [49]. The LIFT-FRRf cuvette was rinsed with deionized water and filtered seawater between sample runs, and blanks were obtained before each set of measurements by quantifying the fluorescence signal in filtered seawater.

For each dark and light acclimated measurement, we applied a Single Turnover (ST) protocol to derive primary photosynthetic parameters. This ST protocol consists of an excitation phase with 100 flashlets of 1.6 μs, with an interval of 2.5 μs between flashlets, followed by relaxation phase consisting of 127 flashlets of 1.6 μs separated by an exponentially increasing interval (initially 20 μs) to allow gradual reoxidation of photosynthetic electron carriers. A total of 5 flashes (each consisting of 32 acquisitions) were run for each sample, and the primary chl-a

fluorescence parameters were derived by fitting the fluorescence transient curves measured at 445 nm with a biophysical model [50] using LIFT software (Soliense Inc.). The first two flashes were typically discarded from the analysis, as well as any data with a signal-to-noise ratio threshold $\leq$10. For most acquisitions, the final three flashes from each run were averaged per replicate, per time point. Using the primary fluorescence parameters, Stern-Volmer non-photochemical quenching ($NPQ_{sv}$) was calculated as:

$$NPQ_{sv} = \frac{F_m - F_m\prime}{F_m\prime} \tag{2}$$

where $F_m$ is the maximum chl-a fluorescence measured in the light (denoted as $F_m\prime$) or in the dark [51]. Maximum photosynthetic efficiency ($F_v/F_m$) was calculated from dark-adapted samples as:

$$F_v/F_m = \frac{F_m - F_o}{F_m} \tag{3}$$

Where $F_o$ is the minimum fluorescence measured in dark acclimated samples.

## Statistics

Student's two-tailed t-tests were used to assess differences in DMSO reduction rate constants between LL and HL treatments for each station. Assumptions of normality and homoscedasticity were validated using Shapiro-Wilks and Levene's tests, respectively. These statistical tests were performed in Python (v.3.8.16) using the "scipy" (v.1.10.1) package. To test for significant differences in $F_v/F_m$ over time between LL and HL treatments, we applied a two-way mixed effects ANOVA, with the assumptions of normality and homoscedasticity validated using Shapiro-Wilks and Levene's tests, respectively. These statistical tests were performed in R (v.4.3.3) using the "lme4" (v.1.1–35.3), "stats" (v.4.3.3) and "car" (v.3.1–2) packages.

## Results and discussion

### Irradiance-driven variability in DMS/P/O cycling

We conducted a series of incubation experiments along the west coast of Vancouver Island (S1 Fig) in the NE Subarctic Pacific (NESAP), to examine changes DMS/P/O cycling in response to experimental light manipulations an area of high DMS concentrations and turnover rates [5,30,52]. Natural plankton assemblages collected from the sub-surface chl-a maximum were spiked with isotopically labelled DMS, DMSP and DMSO, and exposed to either 1% (low light, LL) or 50% (high light, HL) surface irradiance (*i.e.* PAR) to induce different levels of photooxidative stress. All stations showed well developed mixed layers and varying depths of the sub-surface chl-a maximum (Fig 1D), allowing our deck-board incubations to simulate the photo-inhibitive effects of vertical transport on sub-surface assemblages mixed to the surface.

We found that irradiance had no consistent effect on DMSP cleavage rates (S1 Table). Statistically significant, light-dependent differences in D6-DMSP cleavage rates were only observed at station CS04, which showed 50.2% inhibition under HL (t = 12.56, p < 0.01), and station SS2, which showed an opposite response, with an 81.5% increase in DMSP cleavage under HL (t = -6.94, p < 0.01). Previous work has also shown variable responses of DMSP cleavage under high irradiance exposure [53], and this has been attributed to light-dependent inhibition of either bacterial DMSP demand [26,54] or eukaryotic DMSP cleavage [55], depending on the community's taxonomic composition [53]. As with DMSP cleavage, gross DMS consumption, when detectable, was not significantly affected by elevated irradiance

exposure (S1 Table). These DMS consumption rates are believed to primarily reflect DMS uptake and loss through biological oxidation to DMSO ([3], and unpublished NESAP work).

In contrast to DMSP cleavage and DMS consumption, we found that DMSO reduction rate constants increased significantly (up to ~ 3-fold) under high irradiance exposure. Importantly, this increase was only apparent at stations with a chl-a maximum depth greater than 15 m (Fig 1A and 1D), where phytoplankton predominantly experienced low depth-integrated irradiance levels (S2 Fig). Several lines of evidence suggest that the observed light-dependent upregulation in DMSO reduction was influenced by phytoplankton photo-acclimation states. At station LBP8, where phytoplankton were sampled from a relatively shallow, high irradiance chl-a maximum (Figs 1D and S2), biomass-normalized concentrations of xanthophylls and other photoprotection pigments (which were sampled prior to the start of incubations) were 1.5 to 2-fold higher than at station SS2, where the chl-a maximum was deeper and experienced lower irradiance (Figs 1D, 1E and S2). Xanthophyll cycling represents a component of non-photochemical quenching (NPQ), a primary photoprotective mechanism [22,41], and the de-epoxidation ratio of the xanthophylls diatoxanthin to diadinoxanthin (Dt/Dd) has been used as a diagnostic of active NPQ [22,41]. We found that the Dt/Dd ratio was 1.3-fold higher at LBP8 relative to station SS2 (Fig 1E), consistent with high light acclimation of phytoplankton at this station. This conclusion is further supported by the lack of irradiance effects on maximum photosynthetic efficiency ($F_v/F_m$) at station LBP8 (shallow chl-a maximum; Fig 1A and 1B), as compared to a ~33% decrease in $F_v/F_m$ at station SS2 (deeper chl-a maximum) under high irradiance exposure ($F_{(1,16)}$ = 254.97, $p < 0.001$; Fig 1B). Together, these results indicate that phytoplankton sampled from the deeper chl-a maximum at station SS2 were more sensitive to photo-inhibition in high light experimental treatments. In the SS2 assemblage, and most others sampled from deeper low-light waters, DMSO reduction rates increased significantly under high irradiance (Fig 1A).

Further evidence linking DMSO reduction to photo-acclimation comes from the strong inverse correlation observed between DMSO reduction and NPQ (quantified as Stern-Volmer NPQ; r = -0.53 for pooled data) across different sampling stations (Fig 2). This result, along with that of our light manipulation experiments, provides evidence for a metabolic trade-off between different photo-protective mechanisms. Under low-light conditions, phytoplankton downregulate primary photoprotective mechanisms (*e.g.* xanthophyll cycling, NPQ, and photoprotective carotenoids), while upregulating chl-a concentrations to both maximize light harvesting potential and to reduce metabolic costs [29]. However, when these low-light assemblages experience rapid exposure to high irradiance (*e.g.* through vertical mixing events [24]), re-acclimation to reach steady state requires pigment synthesis and dilution of chl-a (through cellular growth and divisions). This process can take hours [24,29], leaving chloroplast machinery susceptible to energy saturation and oxidative damage. In the absence of primary protection mechanisms, rapidly inducible secondary photo-protection mechanisms, including antioxidants or alternative electron pathways [22,56,57], can be upregulated to avoid oxidative stress associated with light shock [29]. We suggest that DMSO reduction can serve as one such secondary photo-protective pathway, when primary protection mechanisms (*e.g.* xanthophyll cycling, NPQ) are down-regulated (Fig 1F). Below, we present evidence to support this hypothesis.

## An electron scavenging mechanism for DMSO

Our experiments point to a link between DMSO reduction and irradiance-induced oxidative stress, presumably caused by the accumulation of ROS. These results, however, do not provide insight into the underlying physiological mechanisms linking DMSO and ROS. Intracellular

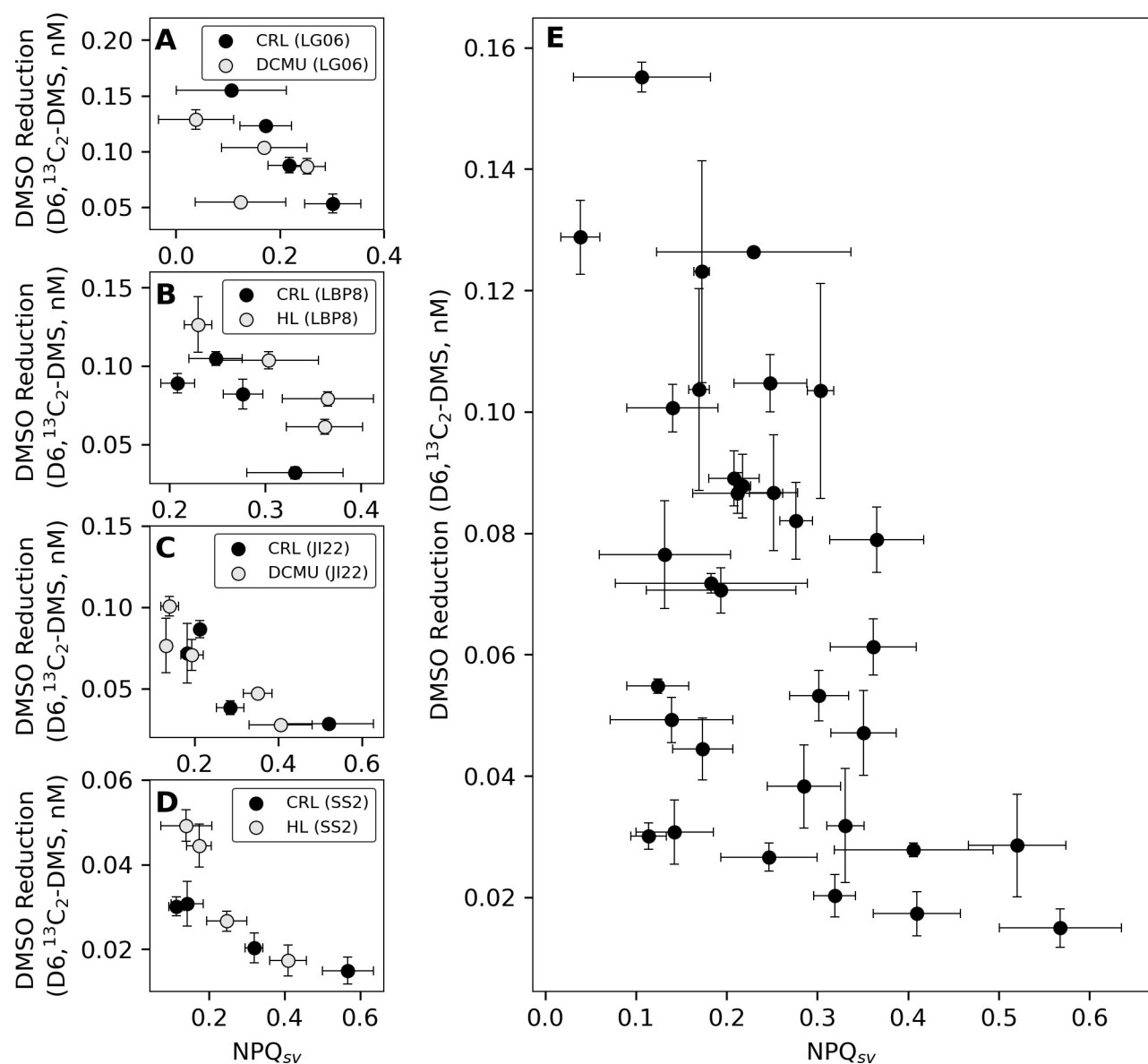

**Fig 2. Correlations between DMSO reduction (D6,$^{13}$C$_2$-DMS production) and Stern-Volmer non-photochemical quenching (NPQ$_{sv}$) at four sampling stations.** Data are plotted for the experimental treatment (either high light or DCMU; grey markers) and their respective control (black markers) for stations **(A)** LGO6, **(B)** LBP8, **(C)** JI22, and **(D)** SS2. **(E)** Pooled data for all three treatments at the four sampling stations illustrate the overall inverse correlation between DMSO reduction and NPQ. Error bars denote range (n = 2) or ± 1 s.d. (n = 3). Except for JI22 (D6,$^{13}$C$_2$-DMS: n = 2), all means are n = 3. Pearson correlations (r) for pooled data is -0.53, and individual treatments range from -0.82 to -0.97, except under DCMU exposure at LG06, where r = -0.41.

ROS production occurs when excess absorbed photons in light-harvesting centers are not sufficiently quenched, leading to either photosystem II (PSII) charge recombination (producing singlet oxygen [22]), or excess production of electrons that can react with O$_2$, particularly at photosystem I (PSI) where a greater fraction of ROS are produced (*e.g.* through dismutation and ascorbate-peroxidase reactions [22,29]; red box in Fig 3D). Oxidative stress can therefore be mitigated by either scavenging ROS through pathways reducing ROS, or by scavenging of excess electrons prior to ROS formation [58–60].

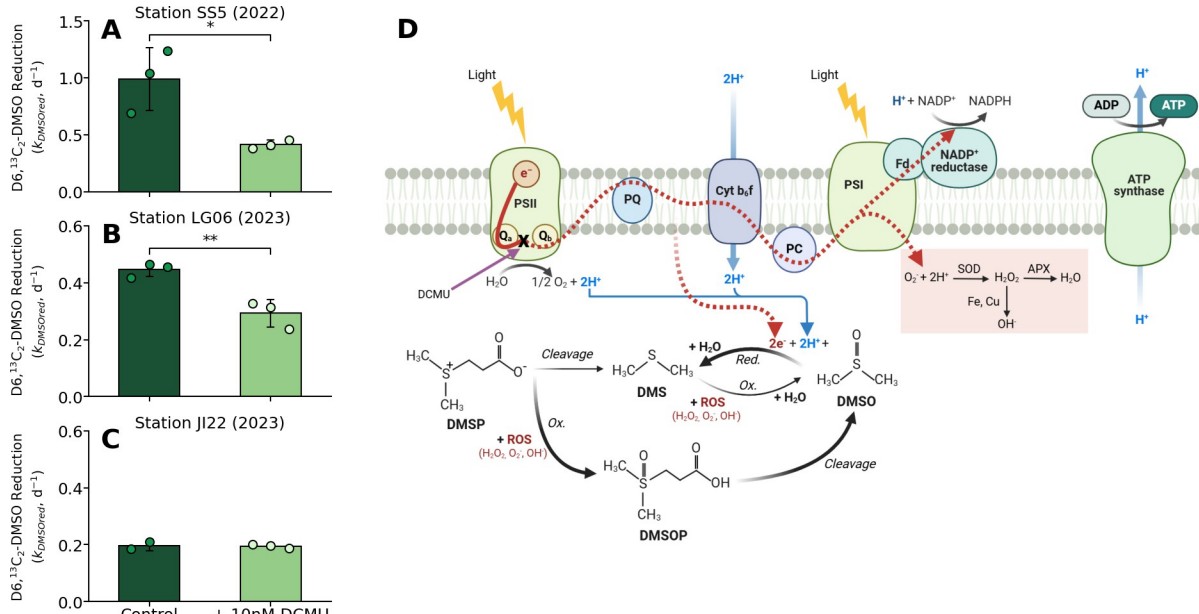

**Fig 3. Proposed mechanistic links between methylated sulfur cycling and photosynthetic electron transport in marine phytoplankton.**
**(A-B)** The effect of exposure to DCMU (which specifically blocks electron flow past the secondary quinone ($Q_b$) in PSII) [61] on D6,$^{13}$C$_2$-DMSO reduction rate constants ($k_{DMSOred}$, in d$^{-1}$; bars) at stations **(A)** SS5 (2022), **(B)** LG06 (2023) and **(C)** JI22 (2023). Data points represent individual replicate measurements, while bars represent the mean value for each treatment. Error bars indicate range (n = 2) or ± 1 s.d. (n = 3). Statistical significance is determined by two-tailed Student's t-tests (*: $p < 0.05$, **: $p < 0.01$). **(D)** A schematic representation of the proposed energy dissipation pathway in which DMSO acts as a sink for excess energy downstream of photosystem II (PSII) that would otherwise produce ROS *e.g.* at photosystem I (PSI; see red box). The acidic lumen, produced from the water splitting reactions at PSII and the trans-thylakoid proton gradient at Cytochrome b$_6$f (Cyt b$_6$f), can satisfy the proton requirements for DMSO reduction. The oxidation of intracellular DMSP [19] and cleavage of the intermediate DMSOP [16] could act to replenish intracellular DMSO concentrations and further mitigate ROS build-up, providing an efficient antioxidant redox system [18,28]. Smaller arrows signify that the DMSP cleavage and DMS oxidation pathways likely provide an indirect, minor contribution towards this hypothetical antioxidant system (see main text) [21]. Electron flow across redox carriers is indicated by the red arrow, with dashes denoting inhibition by DCMU past $Q_b$ (purple arrow). Additional abbreviations in **(D)** are as follows: reduction (Red.), oxidation (Ox.), primary quinones ($Q_a/Q_b$), plastoquinone (PQ) plastocyanin (PC), ferredoxin (Fd), superoxide dismutase (SOD), ascorbate peroxidase (APX). Schematic created in BioRender. McNabb, B. (2024) https://BioRender.com/w07b221.

To date, the only direct antioxidant mechanism proposed for DMSO is a ROS scavenging function, through which DMSO is oxidized to methanesulfinic acid [18]. Other work has also suggested that DMSO reduction could indirectly contribute to ROS scavenging through recycling of DMS pools and their subsequent oxidation [18,28]. Contrary to this proposed pathway, our tracer-based measurements showed no evidence of increased DMS oxidation rates under high irradiance.

Alternatively, DMSO reduction could play a direct antioxidant role by scavenging high energy electrons upstream of ROS production sites through the following enzyme-catalyzed reaction:

$$DMSO + 2e^- + H^+ \rightarrow DMS + H_2O \tag{1}$$

In bacteria, DMSO has been established as a nearly ubiquitous alternate respiratory electron acceptor [62], but the role of DMSO reduction in phytoplankton remains unclear. The putative enzymes catalyzing eukaryotic DMSO reduction, methionine sulfoxide reductases (MSR) [28], are mainly produced in the chloroplast or cytosol [14,63], and models suggest intracellular DMSO production in phytoplankton is localized to the chloroplasts [64]. Such localization

would enable MSR to capture photosynthetically-derived electrons near their source of production.

To examine the potential link between photosynthetically-derived electrons and DMSO reduction, we exposed natural assemblages to 10 nM DCMU to induce partial inhibition of linear electron flow past PSII [61]. Although little is known about the effects of DCMU on MSR expression in marine phytoplankton, this inhibitor has no effect on MSR isozyme expression in *Chlamydomonas reinhardtii* [65], and promotes MSR expression in the macroalgae *Ulva fasciata* [66]. We thus expect that the DCMU treatments primarily reduced the availability of photosynthetically-derived electrons for DMSO reduction, rather than inhibiting the enzymatic efficiency of this pathway.

In experiments conducted on two separate cruises, we found that the rates of net community DMSO reduction were significantly inhibited in the presence of DCMU by ~ 34 and 58% (Fig 3A and 3B). Since our isotopic method quantifies DMSO reduction via the net yield of $D6,^{13}C_2$-DMS, it is possible (although unlikely, as discussed below) that the observed decrease in DMSO reduction resulted from enhanced $D6,^{13}C_2$-DMS re-oxidation in response to DCMU-induced singlet oxygen production [18,61]. We found that D3-DMS oxidation rates were not significantly upregulated at station SS5 (S3 Fig), indicating that DMS re-oxidation could not account for the inhibition in DMSO reduction. This result, combined with the absence of a light-dependent effect on gross DMS consumption rates, provides further evidence that DMS did not serve an intracellular ROS scavenging role in our experiments (Fig 3A–3C). We thus conclude that the inhibition of DMSO reduction resulted from a decrease in electron supply downstream of PSII, supporting the hypothesis that DMSO acts as a secondary photosynthetic electron acceptor downstream of PSII (Fig 3D).

The hypothesized electron scavenging role for DMSO could potentially form part of a larger oxidative redox mechanism [28], complemented by intracellular ROS-scavenging from DMSP [18,19] (Fig 3D). In this scenario, direct DMS re-oxidation to DMSO inside the chloroplast is likely negligible due to rapid diffusion of DMS across membranes, which results in substantially lower intracellular concentrations (<1 nM, normalized to cell volume) [21] that make this molecule unlikely to outcompete other antioxidants (*e.g.* DMSP) in scavenging intracellular ROS [21]. DMS that is not produced from DMSP cleavage is thus likely generated as a byproduct of coupled DMSP-DMSO redox cycling to scavenge both intracellular ROS and electrons [19,21,28]. The recent discovery of the intermediate DMSOP provides a direct pathway for DMSO formation from DMSP [16] (Fig 3D), eliminating the requirement for DMSP cleavage (and subsequent DMS oxidation) to replenish intracellular DMSO reserves. Rates of DMSOP formation and cleavage have not been measured *in situ*, and it is thus difficult to quantify the importance of this molecule in maintaining intracellular DMSO levels. However, unlike DMS and DMSO [21,64], DMSP is membrane-impermeable without zwitterion-selective transport [67], and recent imaging results have demonstrated high DMSP accumulation inside chloroplasts [68]. These findings, combined with the high oxidative potential of DMSP with ROS produced inside chloroplasts [18,56], suggests that intracellular DMSOP formation and cleavage could indeed represent a significant source of DMSO near site of photosynthetically produced electrons, therefore potentially facilitating an efficient ROS and electron scavenging mechanism inside chloroplasts. However, DMSP-DMSO redox coupling may be a less viable strategy for phytoplankton species lacking strong DMSP synthesis capability [67]. For these species, DMSO reduction could still be a useful photoprotective mechanism independent of intracellular DMSP and DMS turnover, as these phytoplankton should be capable of direct DMSO uptake due to both its high membrane permeability [64] and its elevated concentrations in seawater [15,69].

In addition to acting as a potential photo-protective sink for excess photosynthetic electrons, DMSO reduction also consumes protons, and could modulate intracellular pH gradients (ΔpH) that trigger other photo-protective mechanisms. For example, strong ΔpH and lumen acidity is needed for the epoxidation reactions governing PSII xanthophyll cycling [41], which contribute to regulated NPQ thermal dissipation [29]. Proton consumption via DMSO reduction would decrease ΔpH and lumen acidity (Eq 1 and Fig 3D), decreasing cellular capability for the induction of xanthophyll cycling and NPQ. This mechanism could help explain the inverse correlations we observed between DMSO reduction and both xanthophyll cycling and NPQ (Figs 1E, 3 and S4). An analogous positive ΔpH feedback mechanism has been proposed between NPQ and the alternative electron acceptor plasmid terminal oxidase (PTOX) [57], suggesting that a similar physiological role for DMSO is plausible. We note, however, that the extent to which ΔpH exerts regulatory feedback over NPQ in phytoplankton is both debated and taxa-specific, and for some species, ΔpH is restricted to only controlling NPQ activation [41].

Our results suggest that DMSO reduction may act to scavenge excess photosynthetic electrons in some phytoplankton assemblages, but it is unclear how widespread this might be across different taxonomic groups. At station JI22, which was primarily haptophyte-dominated (based on HPLC pigment analysis), DCMU exposure did not significantly inhibit DMSO reduction (Figs 3C and S5). In contrast to station JI22, significant DCMU inhibition of DMSO reduction was observed at station LG06, despite similar hydrographic properties (1-m difference in MLD). Notably, phytoplankton assemblages at station LG06 were somewhat different than those at station JI22, with assemblages dominated by both haptophytes and dinoflagellates, and dinoflagellate abundance was 24% higher than at station JI22 (S5 Fig and Fig 3B). Previous work in the NESAP has demonstrated lower DMSO reduction rates in phytoplankton assemblages dominated by high DMSP-producing haptophytes (as at station JI22), and higher reduction rates in assemblages with a greater proportions of dinoflagellates (as at station LG06) [12]. It is possible that the phytoplankton at station JI22 were not strong DMSO reducers, and were thus unaffected by DCMU exposure. It is also possible that DMSO reduction at this station was driven primarily by respiratory processes in heterotrophic microbes [12,62], and thus independent of photosynthetic electron flow and DCMU inhibition effects. More work is needed to better understand taxonomic variability in DMSO reduction and its potential to serve a photo-protective role. We note that previous work has repeatedly demonstrated taxa-specific sulfur utilization strategies [14,20], which influence the contributions of DMSP and DMSO to net community DMS production.

## Conclusion

We have presented evidence that DMSO serves a photo-protective role in marine phytoplankton through reductive electron scavenging under conditions of excess irradiance. Our findings suggest a mechanism linking irradiance, vertical mixing, and DMS production [25–27] through a stress-driven DMSO reduction pathway. Previous work has shown that wind-driven vertical mixing can stimulate rapid surface DMS production on short-time scales [70], and elevated coastal DMS concentrations have been correlated with upwelling of sub-surface phytoplankton into higher irradiance surface waters [30]. Our results suggest that simulated rapid vertical mixing can increase DMS production via DMSO reduction, as part of an electron scavenging photo-protective mechanism [18,24,29] (Figs 1F and 3D), providing a physiological explanation for the observed correlations between DMS production and vertical mixing [25–27]. The apparent lack of significant light-dependent DMS oxidation rates suggests a limited role for DMS in ROS scavenging in the assemblages we sampled [18]. Rather, DMS may have

accumulated as a byproduct of DMSO reduction [21], or served other physiological roles (*e.g.* cell signalling) [1] in these assemblages.

The light-dependent upregulation of DMSO reduction to DMS is consistent with the negative feedback mechanism proposed by the CLAW hypothesis [6]. Although the validity of this hypothesis has been questioned in recent years [71], there is evidence that DMS influences regional climate forcing, at least seasonally (reviewed in [7]). Irradiance-driven increases in DMSO reduction, and any climatic negative feedback effects from the resulting DMS, might be restricted to short-lived seasonal processes (*e.g.* during upwelling initiation, prior to high-light acclimation) or transient mixing events (*e.g.* rapid deep vertical mixing and subsequent re-stratification following storm activity) [24,70]. Nonetheless, the proposed electron scavenging mechanism for DMSO should extend to other oxidative stressors that influence photosynthetic electron flow (*e.g.* iron or nitrogen limitation) [28,72], which would make our findings potentially applicable across a diverse range of oceanic conditions.

## Supporting information

**S1 Fig. Map of stations sampled during the 2023 and 2022 cruises along the coastal NESAP.** Markers indicate the type of experiments conducted; black circles denote light manipulations, while stars represent DCMU addition experiments. Red labels denote stations sampled in 2023. The dashed contour line indicates the 200 m depth contour, using bathymetry data obtained from US National Geophysical Data Center (NGDC) ETOPO2 dataset (retrieved from: https://rda.ucar.edu/datasets/d759003/). The inset map highlights the study region (red box) in the orthographic projection. These maps were produced using the "Cartopy" package (v0.21.1) in python (v3.8.16), with land features obtained from the Natural Earth dataset retrieved from the public domain (see https://www.naturalearthdata.com/about/).
(TIF)

**S2 Fig. Depth-integrated photosynthetically active radiation (PAR; µE m$^{-2}$ s$^{-1}$) values throughout the euphotic zone at each of the light manipulation experimental stations.** Note the order of stations is the same as Fig 2 in the main text.
(TIF)

**S3 Fig. Oxidation rates of D3-DMS and D6-DMSP under DCMU exposure at station SS5.** **(A)** Hourly yields of D3-DMSO (nM) derived from D3-DMS oxidation for control and 10 nM DCMU treatments. **(B)** Rate constants of D3-DMS oxidation ($k_{DMSox}$, d$^{-1}$) derived from the linear regression of time-course data shown in (A). Data points represent individual replicate measurements. **(C)** same as **(A)**, for D6-DMSO formation from D6-DMSP oxidation. **(D)** same as **(B)** for rate constants of D6-DMSO formation ($k_{DMSPox}$, d$^{-1}$). All error bars indicate ± 1 s.d.
(TIF)

**S4 Fig. Comparison of DMSO reduction rates and the concentration of xanthophyll cycling and photoprotective pigments at stations LBP8 and SS2. (A)** Rate constants D6,$^{13}$C$_2$-DMSO reduction ($k_{DMSOred}$, d$^{-1}$; n = 3) for low light (LL) and high light (HL) treatments. Significance derived from two-tailed Student's t-tests (*: $p < 0.05$, n.s.: non-significant), error bars denote ± 1 s.d. **(B)** Concentration of photoprotective pigments (µg L$^{-1}$) obtained from HPLC analysis and calculated as the sum total of 9-cis-neoxanthin, violaxanthin, diadinoxanthin, diatoxanthin, alloxanthin, lutein, zeaxanthin, and β-carotene concentrations. **(C)** Concentrations of diadinoxanthin (Dd, µg L$^{-1}$; light bars) and diatoxanthin (Dt, µg L$^{-1}$; grey bars) normalized to total chlorophyll-a (chl-a, µg L$^{-1}$) concentrations. Blue markers indicate the de-epoxidation ratios of the absolute concentrations of Dd and Dt. **(D)** Same as **(C)** for

concentrations of zeaxanthin (Zea, µg L$^{-1}$; light bars) and violaxanthin (Viol, µg L$^{-1}$; grey bars) normalized to total chl-a, and their epoxidation ratios (blue markers). All error bars indicate range (n = 2) for pigment data.
(TIF)

**S5 Fig. Comparison of phytoplankton assemblage composition for stations with different DMSO reduction responses to DCMU treatment. (A)** Relative abundances (%) of different phytoplankton groups at stations JI22 and LG06 derived from CHEMTAX analysis. **(B)** Rate constants of D6,$^{13}$C$_2$-DMSO reduction ($k_{\text{DMSOred}}$, d$^{-1}$) in the control and +10 nM DCMU treatments for stations JI22 and LG06. Significance derived from two-tailed Student's t-tests (**: $p < 0.01$, n.s.: non-significant). Error bars indicate range (n = 2) or ± 1 s.d. (n = 3).
(TIF)

**S1 Table. Experimental DMS turnover rate constants, $k$ (d$^{-1}$), in the coastal NE Subarctic Pacific.**
(DOCX)

**S2 Table. Data availability (indicated by 'x') for this study.**
(DOCX)

## Acknowledgments

We thank Ross McCulloch, Akash Sastri, John Nelson, and the crew of the *CCGS John P. Tully* for logistical and technical support, Nicole Link and Oliver Pankratz for assistance in data collection, and James Pinckney for analysis of HPLC samples.

## Author Contributions

**Conceptualization:** Brandon J. McNabb, Philippe D. Tortell.

**Data curation:** Brandon J. McNabb.

**Formal analysis:** Brandon J. McNabb.

**Funding acquisition:** Brandon J. McNabb, Philippe D. Tortell.

**Investigation:** Brandon J. McNabb.

**Methodology:** Brandon J. McNabb, Philippe D. Tortell.

**Project administration:** Philippe D. Tortell.

**Resources:** Philippe D. Tortell.

**Supervision:** Philippe D. Tortell.

**Validation:** Brandon J. McNabb, Philippe D. Tortell.

**Visualization:** Brandon J. McNabb.

**Writing – original draft:** Brandon J. McNabb.

**Writing – review & editing:** Philippe D. Tortell.

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
