## [Decision Letter · Decision Letter 0]

11 Nov 2024

PONE-D-24-45077A photo-protective antioxidant function for DMSO in marine phytoplankton.PLOS ONE

Dear Dr. McNabb,

Thank you for submitting your manuscript to PLOS ONE. After careful consideration, we feel that it has merit but does not fully meet PLOS ONE’s publication criteria as it currently stands. Therefore, we invite you to submit a revised version of the manuscript that addresses the points raised during the review process.

We look forward to receiving your revised manuscript.

Kind regards,

Thomas Roach

Academic Editor

PLOS ONE

Journal Requirements:

2. Thank you for stating the following financial disclosure: “This work was supported by grants awarded to P.D.T. and B.J.M through the National Science and Engineering Research Council (NSERC).”

4. We note that Supporting Information Figure 1 in your submission contain [map/satellite] images which may be copyrighted. All PLOS content is published under the Creative Commons Attribution License (CC BY 4.0), which means that the manuscript, images, and Supporting Information files will be freely available online, and any third party is permitted to access, download, copy, distribute, and use these materials in any way, even commercially, with proper attribution. For these reasons, we cannot publish previously copyrighted maps or satellite images created using proprietary data, such as Google software (Google Maps, Street View, and Earth). For more information, see our copyright guidelines: http://journals.plos.org/plosone/s/licenses-and-copyright. We require you to either (a) present written permission from the copyright holder to publish these figures specifically under the CC BY 4.0 license, or (2) remove the figures from your submission:

a. You may seek permission from the original copyright holder of Supporting Information Figure 1 to publish the content specifically under the CC BY 4.0 license. We recommend that you contact the original copyright holder with the Content Permission Form (http://journals.plos.org/plosone/s/file?id=7c09/content-permission-form.pdf) and the following text: “I request permission for the open-access journal PLOS ONE to publish XXX under the Creative Commons Attribution License (CCAL) CC BY 4.0 (http://creativecommons.org/licenses/by/4.0/). Please be aware that this license allows unrestricted use and distribution, even commercially, by third parties. Please reply and provide explicit written permission to publish XXX under a CC BY license and complete the attached form.” Please upload the completed Content Permission Form or other proof of granted permissions as an "Other" file with your submission. In the figure caption of the copyrighted figure, please include the following text: “Reprinted from [ref] under a CC BY license, with permission from [name of publisher], original copyright [original copyright year].”

b. If you are unable to obtain permission from the original copyright holder to publish these figures under the CC BY 4.0 license or if the copyright holder’s requirements are incompatible with the CC BY 4.0 license, please either i) remove the figure or ii) supply a replacement figure that complies with the CC BY 4.0 license. Please check copyright information on all replacement figures and update the figure caption with source information. If applicable, please specify in the figure caption text when a figure is similar but not identical to the original image and is therefore for illustrative purposes only. The following resources for replacing copyrighted map figures may be helpful: USGS National Map Viewer (public domain): http://viewer.nationalmap.gov/viewer/ The Gateway to Astronaut Photography of Earth (public domain): http://eol.jsc.nasa.gov/sseop/clickmap/ Maps at the CIA (public domain): https://www.cia.gov/library/publications/the-world-factbook/index.html and https://www.cia.gov/library/publications/cia-maps-publications/index.html NASA Earth Observatory (public domain): http://earthobservatory.nasa.gov/ Landsat: http://landsat.visibleearth.nasa.gov/ USGS EROS (Earth Resources Observatory and Science (EROS) Center) (public domain): http://eros.usgs.gov/# Natural Earth (public domain): http://www.naturalearthdata.com/

Reviewers' comments:

Reviewer's Responses to Questions

**Comments to the Author**

1. Is the manuscript technically sound, and do the data support the conclusions?

Reviewer #1: Yes

Reviewer #2: Yes

2. Has the statistical analysis been performed appropriately and rigorously? 

Reviewer #1: Yes

Reviewer #2: Yes

3. Have the authors made all data underlying the findings in their manuscript fully available?

Reviewer #1: Yes

Reviewer #2: Yes

4. Is the manuscript presented in an intelligible fashion and written in standard English?

Reviewer #1: Yes

Reviewer #2: Yes

5. Review Comments to the Author

Reviewer #1: I have reviewed the above manuscript submitted by McNabb and Tortell regarding experiments conducted on a cruise that examined the response of DMSO reduction with respect to both light treatments and DCMU application. Their conclusions are that DMSO scavenges excess electrons coming from PSII in a secondary photoprotective response. I find that the results presented directly address the hypothesis and support it strongly. Their conclusions are not too speculative and they clearly articulate potential confounding factors. Their work is timely and provides new and interesting light on the DMS/O/P system in marine phytoplankton.

Reviewer #2: please see attached document

6. PLOS authors have the option to publish the peer review history of their article (what does this mean?). If published, this will include your full peer review and any attached files.

Reviewer #1: No

Reviewer #2: No

---

## [Author Response · Author response to Decision Letter 0]

12 Dec 2024

We have attached a rebuttal letter that includes line-by-line changes to address the comments and concerns from both reviewers.

---

## [Decision Letter · Decision Letter 1]

2 Jan 2025

PONE-D-24-45077R1A potential photo-protective, antioxidant function for DMSO in marine phytoplankton.PLOS ONE

Dear Dr. McNabb,

Thank you for submitting your manuscript to PLOS ONE. Reviewer 1 has raised a valid point about the permeability of the membrane to DMSO, which seems contradictory in your submission and also found a typo. If these points are revised I can accept your manuscript immediately.

We look forward to receiving your revised manuscript.

Kind regards,

Thomas Roach

Academic Editor

PLOS ONE

Journal Requirements:

Reviewers' comments:

Reviewer's Responses to Questions

**Comments to the Author**

1. If the authors have adequately addressed your comments raised in a previous round of review and you feel that this manuscript is now acceptable for publication, you may indicate that here to bypass the “Comments to the Author” section, enter your conflict of interest statement in the “Confidential to Editor” section, and submit your "Accept" recommendation.

Reviewer #1: All comments have been addressed

Reviewer #2: All comments have been addressed

2. Is the manuscript technically sound, and do the data support the conclusions?

Reviewer #1: Yes

Reviewer #2: (No Response)

3. Has the statistical analysis been performed appropriately and rigorously? 

Reviewer #1: Yes

Reviewer #2: (No Response)

4. Have the authors made all data underlying the findings in their manuscript fully available?

Reviewer #1: Yes

Reviewer #2: (No Response)

5. Is the manuscript presented in an intelligible fashion and written in standard English?

Reviewer #1: Yes

Reviewer #2: (No Response)

6. Review Comments to the Author

Reviewer #1: I have read the above manuscript and find it much improved from the original draft. I do, however, still have some comments to make – one minor and one more involved.

The minor comment is an apparent typo on L199, “diadinoxanthin (Dt) and diadinoxanthin (Dd)”. Later, on L298, Dt is reported as “diatoxanthin”.

The more involved comment regards the paragraph on L413-437. Beginning on L416, the authors argue that DMSO has sufficient polarity to restrict diffusive loss across the membrane, citing two studies. Unfortunately, this statement is not supported by those studies or many others. DMSO rapidly crosses cell membranes. Lavoie et al. [ref. 64] have an extensive discussion about the factors that may impact their model, particularly concerning the multiple membranes in the chloroplast, if that is where DMSO is located. Indeed, even the authors indicate a high membrane permeability for DMSO on L432.

In this same paragraph, I would suggest to the authors to be clearer about DMSP compartmentalization. According to ref. 67, DMSP is localized to the chloroplast, as indicated on L428. Unlike DMSO, DMSP is membrane-impermeable owing to its zwitterionic nature. This would suggest that oxidation to DMSO(P) is actually highly likely, providing a DMSO source in situ without the need to argue for uptake from bulk seawater (L433), unless (as the authors state) the phytoplankton in question lack significant DMSP production.

Reviewer #2: (No Response)

7. PLOS authors have the option to publish the peer review history of their article (what does this mean?). If published, this will include your full peer review and any attached files.

Reviewer #1: No

Reviewer #2: No

---

## [Author Response · Author response to Decision Letter 1]

6 Jan 2025

We thank you the reviewer for their additional comments and have enclosed a response letter outlining the requested changes made.

---

## [Editor Report · Decision Letter 2]

7 Jan 2025

A potential photo-protective, antioxidant function for DMSO in marine phytoplankton.

PONE-D-24-45077R2

Dear Dr. McNabb,

We’re pleased to inform you that your manuscript has been judged scientifically suitable for publication and will be formally accepted for publication once it meets all outstanding technical requirements.

Kind regards,

Thomas Roach

Academic Editor

PLOS ONE
---

## [Editor Report · Acceptance letter]

24 Jan 2025

PONE-D-24-45077R2 

PLOS ONE

Dear Dr. McNabb, 

I'm pleased to inform you that your manuscript has been deemed suitable for publication in PLOS ONE. Congratulations! Your manuscript is now being handed over to our production team.

Kind regards, 

on behalf of

Dr. Thomas Roach 

Academic Editor

PLOS ONE